# Recursive Reinforcement Learning

**Ernst Moritz Hahn**[*] 
University of Twente

**Mateo Perez**[*] 
University of Colorado Boulder

**Sven Schewe**[*] 
University of Liverpool

**Fabio Somenzi**[*] 
University of Colorado Boulder

**Ashutosh Trivedi**[*] 
University of Colorado Boulder

**Dominik Wojtczak**[*] 
University of Liverpool

## Abstract

Recursion is the fundamental paradigm to finitely describe potentially infinite objects. As state-of-the-art reinforcement learning (RL) algorithms cannot directly reason about recursion, they must rely on the practitioner's ingenuity in designing a suitable "flat" representation of the environment. The resulting manual feature constructions and approximations are cumbersome and error-prone; their lack of transparency hampers scalability. To overcome these challenges, we develop RL algorithms capable of computing optimal policies in environments described as a collection of Markov decision processes (MDPs) that can recursively invoke one another. Each constituent MDP is characterized by several entry and exit points that correspond to input and output values of these invocations. These recursive MDPs (or RMDPs) are expressively equivalent to probabilistic pushdown systems (with call-stack playing the role of the pushdown stack), and can model probabilistic programs with recursive procedural calls. We introduce *Recursive Q-learning*— a model-free RL algorithm for RMDPs—and prove that it converges for finite, single-exit and deterministic multi-exit RMDPs under mild assumptions.

## 1 Introduction

Reinforcement learning [36] (RL) is a stochastic approximation based approach to optimization, where learning agents rely on scalar reward signals from the environment to converge to an optimal behavior. Watkins's seminal approach [41] to RL, known as Q-learning, judiciously combines exploration/exploitation with dynamic programming to provide guaranteed convergence [40] to optimal behaviors in environments modeled as Markov decision processes (MDPs) with finite state and action spaces. RL has also been applied to MDPs with uncountable state and action spaces, although convergence guarantees for such environments require strong regularity assumptions. Modern variants of Q-learning (and other tabular RL algorithms) harness the universal approximability and ease-of-training rendered by deep neural networks [18] to discover creative solutions to problems traditionally considered beyond the reach of AI [30, 39, 33].

These RL algorithms are designed with a flat Markovian view of the environment in the form of a "state, action, reward, and next state" interface [9] in every interaction with the learning agent, where the states/actions may come from infinite sets. When such infinitude presents itself in the form of finitely represented recursive structures, the inability of the RL algorithms to handle structured environments means that the structure present in the environment is not available to the RL algorithm to generalize its learning upon. The work of [41] already provides a roadmap for hierarchically structured environments; since then, considerable progress has been made in developing algorithms for hierarchical RL [6, 13, 37, 31] with varying optimality guarantees. Still, the hierarchical MDPs

---

[*]Authors listed alphabetically.

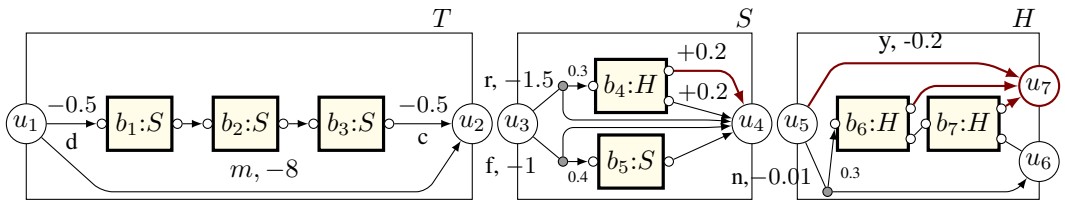

Figure 1: A recursive Markov decision process with three components $T$, $S$, and $H$.

are expressively equivalent to finite-state MDPs, although they may be exponentially more succinct (Lemma 1). Thus, hierarchical RL algorithms are inapplicable in the presence of unbounded recursion.

On the other hand, recursion occurs naturally in human reasoning [10], mathematics and computation [34, 35], and physical environments [29]. Recursion is a powerful cognitive tool in enabling a divide-and-conquer strategy [11] to problem solving (e.g., tower of Hanoi, depth-first search) and, consequently, recursive solutions enhance explainability in the form of intuitive inductive proofs of correctness. Unlike flat representations, the structure exposed by recursive definitions enables generalizability. Recursive concepts, such as recursive functions and data structures, provide scaffolding for efficient and transparent algorithms. Finally, the models of physical environments express the system evolution in the form of recursive equations. We posit that the lack of RL algorithms in handling recursion is an obstacle to their applicability, explainability, and generalizability. This paper aims to fill the gap by studying *recursive Markov decision processes* [16] as environment models in reinforcement learning. We dub this setting recursive reinforcement learning (RRL).

**MDPs with Recursion.**  A recursive Markov Decision Process (RMDP) [16] is a finite collection of special MDPs, called component MDPs, with special entry and exit nodes that take the role of input parameter and return value, respectively. The states of component MDPs may either be the standard MDP states, or they may be "boxes" with input and output ports; these boxes are mapped to other component MDPs (possibly, the component itself) with some matching of the entry and exit nodes. An RMDP where every component has only one exit is called a 1-exit RMDP, otherwise we call it a general or multi-exit RMDP. Single-exit RMDPs are strictly less expressive than general RMDPs as they are equivalent to functions without any return value. Nonetheless, 1-exit RMDPs are more expressive than finite-state MDPs [32] and relate closely to controlled branching processes [17].

**Example 1** (Cloud Computing). *As an example of recursive MDP, consider the Boolean program shown in Figure 2. This example (inspired from [21]) models a cloud computing scenario to compute a task $T$ depicted as the component $T$. Here, a decision maker is faced with two choices: either she can choose to execute the task monolithically (with a cost of 8 units) or chose to decompose the task into three $S$ tasks. The process of decomposition and later combining the results cost 0.5 units. Each task $S$ can either be executed on a fast, but unreliable server that costs 1 unit, but with probability 0.4 the server may crash, and require a recomputation of the task. When executed on a reliable server, the task $S$ costs 1.5 units, however the task may be interrupted by a higher-priority task and the decision maker will be compensated for this delay by earning a 0.2 unit reward. During the interrupt service routine $H$, there is a choice to upgrade the priority of the task for a cost of 0.2 units. Otherwise, the interrupt service routing costs 0.01 unit (due to added delay) and the interrupt service routine itself can be interrupted, causing the service routine to be re-executed in addition to that of the new interrupt service routine. The goal of the RL agent is to compute an optimal policy that maximize the total reward to termination.*

*This example is represented as a recursive MDP in Figure 1. This RMDP has three components $T$, $S$, and $H$. The component $T$ has three boxes $b_1$, $b_2$, and $b_3$ all mapped to components $S$. The component $T$ and $S$ both have single entry and exit nodes, while the component $H$ has two exit nodes. Removing the thick (maroon) transitions and the exit $u_7$ makes the RMDP 1-exit. The edges specify both, the name of the action and the corresponding reward. While the component $T$ is non-stochastic, components $S$ and $H$ both have stochastic transitions depicted by the grey circle.*

Recursive MDPs strictly generalize finite-state MDPs and hierarchical MDPs, and semantically encode countable state MDPs with the context encoding the stack frame of unbounded depth. RMDPs generalize several well-known stochastic models including stochastic context-free grammars [28, 26] and multi-type branching processes [22, 38]. Moreover, RMDPs are expressively equivalent to proba-

```
def T():                      def S():                       def H():
    a = ?({mono, divide});       a = ?({reliable, fast});       upgrade = ?({yes, no});
    if a = mono:                 if a = fast:                   if upgrade = no:
        execute_mono() # −$8       try:                           try:
    else :                             execute_fast() # −$1           ISR() # −$0.01
        decompose() # −$0.5         except Crash: # prob = 0.4     except Interrupt: #prob = 0.3
        S()                            S()                            if (H()):
        S()                      else:                                   return 1
        S()                         try:                            if (H())
        combine() # −$0.5              execute_reliable() #−$1.5       return 1
    return                         except Interrupt: #prob = 0.3   return 0
                                       H() # +$0.2              else : #−$0.2
        (a) Task T                  return                          return 1
                                       (b) Task S                      (c) Task H
```

Figure 2: A probabilistic Boolean program sketch where the choice of the hole (?) is to be filled by RL agent.

bilistic pushdown systems (MDPs with unbounded stacks), and can model probabilistic programs with unrestricted recursion. Given their expressive power, it is not surprising that reachability and termination problems for general RMDPs are undecidable. Single-exit RMDPs, on the other hand, are considerably more well-behaved with decidable termination [16] and total reward optimization under positive reward restriction [15]. Exploiting these properties, a convergent RL algorithm [21] has been proposed for 1-exit RMDPs with positive cost restriction. However, to the best of our knowledge, no RL algorithm exists for general RMDPs.

**Applications of Recursive RL.** Next, we present some paradigmatic applications of recursive RL.

- **Probabilistic Program Synthesis.** As shown in Example 1, RMDPs can model procedural probabilistic Boolean program. Hence, the recursive RL can be used for program synthesis in unknown, uncertain environments. Boolean abstractions of programs [4] are particularly suited to modeling as RMDPs. Potential applications include program verification [4, 5, 14] and program synthesis [19].

- **Context-Free Reward Machines.** Recently, reward machines [23] have been proposed to model non-Markovian reward signals in RL. In this setting, a regular language extended with the reward signals (Mealy machines) over the observation sequences of the MDP is used to encode reward signals. In this setting the RL algorithms operate on the finite MDP given by the product of the MDP with the reward machine. Following the Chomsky hierarchy, context-free grammars or pushdown automata can be used to provide more expressive reward schemes than regular languages. This results in context-free reward machines: reward machines with an unbounded stack. As an example of such a more expressive reward language, consider a grid-world with a reachability objective with some designated charging stations, where 1-unit dwell-time charges the unbounded capacity battery by 1-unit. If every action discharges the battery by 1-unit, the reward scheme to reach the target location without ever draining the battery cannot be captured by a regular reward machine. On the other hand, this reward signal can be captured with an RMDP, where charging by 1-unit amounts to calling a component and discharging amounts to returning from the component such that the length of the call stack encodes the battery level. More generally, any context-free requirement over finite-state MDPs can be captured using general RMDPs.

- **Stochastic Context-Free Grammars.** Stochastic CFGs and branching decision processes can capture a structured evolution of a system. These can be used for modeling disease spread, population dynamics, and natural languages. RRL can be used to learn optimal behavior in systems expressed using such stochastic grammars.

**Overview.** We begin the technical presentation by providing the formal definition of RMDPs and the total reward problem: which we show to be undecidable in general. We then develop PAC learning results under mild restrictions. In Section 3, we develop Recursive Q-learning, a model-free RL algorithm for RMDPs. In Section 4, we show that Recursive Q-learning converges to an optimal solution in the single-exit setting. Section 5 then demonstrates the empirical performance of Recursive Q-learning.

## 2 Recursive Markov Decision Processes

A Markov decision process $\mathcal{M}$ is a tuple $(A, S, T, r)$ where $A$ is the set of *actions*, $S$ is the set of states, $T : S \times A \to \mathcal{D}(S)$ is the probabilistic transition function, and $r : S \times A \to \mathbb{R}$ is the reward function. We say that an MDP $\mathcal{M}$ is finite if both $S$ and $A$ are finite. For any state $s \in S$, $A(s)$ denotes the set of actions that may be selected in state $s$.

A recursive Markov decision process (RMDP) [16] is a tuple $M = (M_1, \ldots, M_k)$, where each *component* $M_i = (A_i, N_i, B_i, Y_i, En_i, Ex_i, \delta_i)$ consists of:

- A set $A_i$ of actions;
- A set $N_i$ of *nodes*, with a distinguished subset $En_i$ of *entry* nodes and a (disjoint) subset $Ex_i$ of *exit* nodes (we assume an arbitrary but fixed ordering on $Ex_i$ and $En_i$);
- A set $B_i$ of *boxes* along with a mapping $Y_i : B_i \mapsto \{1, \ldots, k\}$ that assigns to every box (the index of) a component. To each box $b \in B_i$, we associate a set of *call ports*, $\text{Call}_b = \{(b, en) \mid en \in En_{Y(b)}\}$, and a set of *return ports*, $\text{Ret}_b = \{(b, ex) \mid ex \in Ex_{Y(b)}\}$;
- we let $\text{Call}^i = \cup_{b \in B_i} \text{Call}_b$, $\text{Ret}^i = \cup_{b \in B_i} \text{Ret}_b$, and let $Q_i = N_i \cup \text{Call}^i \cup \text{Ret}^i$ be the set of all nodes, call ports and return ports; we refer to these as the *vertices* of component $M_i$.
- A transition function $\delta_i : Q_i \times A_i \to \mathcal{D}(Q_i)$, where, for each tuple $\delta_i(u, a)(v) = p$ is the transition probability of a transition from the source $u \in (N_i \setminus Ex_i) \cup \text{Ret}^i$ to the destination $v \in (N_i \setminus En_i) \cup \text{Call}^i$; we often write $p(v|u, a)$ for $\delta_i(u, a)(v)$.
- A reward function $r_i : Q_i \times A_i \to \mathbb{R}$ is the reward associated with transitions.

We assume that the set of boxes $B_1, \ldots, B_k$ and set of nodes $N_1, N_2, \ldots, N_k$ are mutually disjoint. We use symbols $N, B, A, Q, En, Ex, \delta$ to denote the union of the corresponding symbols over all components. We say that an RMDP is finite if $k$ and all $A_i$, $N_i$ and $B_i$ are finite.

An execution of an RMDP begins at an entry node of some component and, depending upon the sequence of input actions, the state evolves naturally like an MDP according to the transition distributions. However, when the execution reaches an entry port of a box, this box is stored on a stack of pending calls, and the execution continues naturally from the corresponding entry node of the component mapped to that box. When an exit node of a component is encountered, and if the stack of pending calls is empty then the run terminates; otherwise, it pops the box from the top of the stack and jumps to the exit port of the just popped box corresponding to the just reached exit of the component. The semantics of an RMDP is an infinite state MDP, whose states are pairs consisting of a sequence of boxes, called the context, mimicking the stack of pending calls and the current vertex.

The height of the call stack is incremented (decremented) when a call (return) is made. A stack height of 0 refers to termination, while the empty stack has height 1.

The semantics of a recursive MDP $M = (M_1, \ldots, M_k)$ with $M_i = (A_i, N_i, B_i, Y_i, En_i, Ex_i, \delta_i, r_i)$ are given as a (infinite-state) MDP $[\![M]\!] = (A_M, S_M, T_M, r_M)$ where

- $A_M = \cup_{i=1}^k A_i$ is the set of actions;
- $S_M \subseteq B^* \times Q$ is the set of states, consisting of the stack context and the current node;
- $T_M : S_M \times A_M \to \mathcal{D}(S_M)$ is the transition function such that for $s = (\langle \kappa \rangle, q) \in S_M$ and action $a \in A_M$, the distribution $\delta_M(s, a)$ is defined as:
  1. if the vertex $q$ is a call port, i.e. $q = (b, en) \in \text{Call}$, then $\delta_M(s, a)(\langle \kappa, b \rangle, en) = 1$;
  2. if the vertex $q$ is an exit node, i.e. $q = ex \in Ex$, then if $\kappa = \langle \emptyset \rangle$ then the process terminates and otherwise $\delta_M(s, a)(\langle \kappa' \rangle, (b, ex)) = 1$ where $(b, ex) \in \text{Ret}(b)$ and $\kappa = \langle \kappa', b \rangle$;
  3. otherwise, $\delta_M(s, a)(\langle \kappa \rangle, q') = \delta(q, a)(q')$.
- the reward function $r_M : S_M \times A_M \to \mathbb{R}$ is such that for $s = (\langle \kappa \rangle, q) \in S_M$ and action $a \in A_M$, the reward $r_M(s, a)$ is zero if $q$ is either a call port or the exit node, and otherwise $r_M(s, a)(\langle \kappa \rangle, q') = r(q, a)(q')$. We call the maximum value of the absolute one-step reward the *diameter* of an RMDP and denote it by $r_{max} = \max_{s,a} |r(s, a)|$.

Given the semantics of an RMDP $M$ as an (infinite) MDP $[\![M]\!]$, the concepts of strategies (also called policies) as well as positional strategies are well defined. We distinguish a special class of strategies—called *stackless strategies*—that are deterministic and do not depend on the history or the stack context at all.

We are interested in computing strategies $\sigma$ that maximize the *expected total reward*. Given RMDP $M$, a strategy $\sigma$ determines sequences $X_i$ and $Y_i$ of random variables denoting the $i^{th}$ state and action of the MDP $[\![M]\!]$. The total reward under strategy $\sigma$ and its optimal value are respectively defined as

$$\text{ETotal}_\sigma^M(s) = \lim_{N \to \infty} \mathbb{E}_\sigma^M(s)\Big\{ \sum\nolimits_{1 \le i \le N} r(X_{i-1}, Y_i) \Big\}, \quad \text{ETotal}^M(s) = \sup_\sigma \text{ETotal}_\sigma^M(s).$$

For an RMDP $M$ and a state $s$, a strategy $\sigma$ is called *proper* if the expected number of steps taken by $M$ before termination when starting at $s$ is finite. To ensure that the limit above exists, as the sum of rewards can otherwise oscillate arbitrarily, we assume the following.

**Assumption 1** (Proper Policy Assumption). *All strategies are proper for all states.*

We call an RMDP that satisfies Assumption 1 a *proper RMDP*. This assumption is akin to proper policy assumptions [7] often posed on the stochastic shortest path problems, and ensures that the total expected reward is finite. The expected total reward optimization problem over proper RMDPs subsumes the discounted optimization problem over finite-state MDPs since discounting with a factor $\lambda$ is analogous to terminating with probability $1-\lambda$ at every step [36]. The properness assumption on RMDPs can be enforced by introducing an appropriate discounting (see Appendix F).

**Undecidability.** Given an RMDP $M$, an initial node $v$, and a threshold $D$, the *strategy existence problem* is to decide whether there exists a strategy in $[\![M]\!]$ with value greater than or equal to $D$ when starting at the initial state $(\langle \emptyset \rangle, q)$, i.e., at some entry node $q$ with an empty context.

**Theorem 1** (Undecidability of the Strategy Existence Problem). *Given a proper RMDP and a threshold $D$, deciding whether there exists a strategy with expected value greater than $D$ is undecidable.*

**PAC-learnability.** Although it is undecidable to determine whether or not a strategy can exceed some threshold in a proper RMDP, the problem of $\varepsilon$-approximating the optimal value is decidable when parameters $c_o$, $\lambda$ and $b$ (defined below) are known. Our approach to PAC-learnability [1] is to learn the distribution of the transition function $\delta$ well enough and then produce an approximate, but not necessarily efficient, evaluation of our learned model.

To allow PAC-learnability, we need a further nuanced notion of $\varepsilon$-proper policies. A policy is called $\varepsilon$-proper, if it terminates with a uniform bound on the expected number of steps for all $\mathcal{M}'$ that differ from $\mathcal{M}$ only in the transition function, where $\sum_{q \in S, a \in A, r \in S} |\delta_{\mathcal{M}}(q,a)(r) - \delta_{\mathcal{M}'}(q,a)(r)| \le \varepsilon$ (we then say that $\mathcal{M}'$ is $\varepsilon$-close to $\mathcal{M}$), and where the support of $\delta_{M'}(q,a)$ is a subset of the support of $\delta_M(q,a)$ for all $q \in S$ and $a \in A$. An RMDP is called $\varepsilon$-proper, if all strategies are $\varepsilon$-proper for $M$ for all states of the RMDP.

**Assumption 2** (PAC-learnability). *We restrict our attention to $\varepsilon$-proper RMDPs. We further require that all policies have a falling expected stack height. Namely, we require for all $\mathcal{M}'$ $\varepsilon$-close to $\mathcal{M}$ and all policies $\sigma$ that the expected stack height in step $k$ is bounded by some function $c_o - \mu \cdot \sum_{i=1}^{k} p_{\mathsf{run}\sigma}^{\mathcal{M}'}(k)$, where $c_o \ge 1$ is an offset, $\mu \in ]0,1]$ is the decline per step, and $p_{\mathsf{run}\sigma}^{\mathcal{M}'}(k)$ is the likelihood that the RMDP $\mathcal{M}'$ with strategy $\sigma$ is still running after $k$ steps. We finally require that the absolute expected value from every strategy is bounded: $\left| \text{ETotal}_\sigma^{\mathcal{M}'}((\langle \emptyset \rangle, q)) \right| \le b$ for some $b$.*

**Theorem 2.** *For every $\varepsilon$-proper RMDP with parameters $c_o$, $\mu$, and $b$, $\text{ETotal}^{\mathcal{M}}(s)$ is PAC-learnable.*

These parameters can be replaced by discounting. Indeed, our proofs start with discounted rewards, and then relax the assumptions to allow for using undiscounted rewards. Using a discount factor $\lambda$ translates to parameters $b = \frac{d}{1-\lambda}$, $c_o = 1 + \frac{1}{1-\lambda}$, and $\mu = 1-\lambda$.

## 3 Recursive Q-Learning for Multi-Exit RMDPs

While RMDPs with multiple exits come with undecidability results, they are the interesting cases as they represent systems with an arbitrary call stack. We suggest an abstraction that turns them into a fixed size data structure, which is well suited for neural networks.

Given a proper recursive MDP $M = (M_1, \ldots, M_k)$ with $M_i = (A_i, N_i, B_i, Y_i, En_i, Ex_i, \delta_i, r_i)$ with semantics $[\![M]\!] = (A_M, S_M, T_M, r_M)$, the optimal total expected reward can be captured by the following equations $\text{OPT}_{\text{recur}}(M)$. For every $\kappa \in B^*$ and $q \in Q$:

$$
y(\langle \kappa \rangle, q) = \begin{cases}
y(\langle \kappa, b \rangle, en) & q = (b, en) \in \text{Call} \\
0 & q \in Ex, \kappa = \langle \emptyset \rangle \\
y(\langle \kappa' \rangle, (b, q)) & q \in Ex, (b, q) \in \text{Ret}(b), \kappa = \langle \kappa', b \rangle \\
\max_{a \in A(q)} \left\{ r(q, a) + \sum_{q' \in Q} p(q'|q, a) y(\langle \kappa \rangle, q') \right\} & \text{otherwise.}
\end{cases}
$$

These equations capture the optimality equations on the underlying infinite MDP $[\![M]\!]$. It is straightforward to see that, if these equations admit a solution, then the solution equals the optimal total expected reward [32]. Moreover, an optimal policy can be computed by choosing the actions that maximize the right-hand-side. However, since the state space is countably infinite and has an intricate structure, an algorithm to compute a solution to these equations is not immediate.

To make it accessible to learning, we *abstract* the call stack $\langle \kappa, b \rangle$ to its *exit value*, i.e. the total expected reward from the exit nodes of the box $b$, under the stack context $\langle \kappa \rangle$. Note that when a box is called, the value of each of its exits may still be unknown, but it is (for a given strategy) fixed. Naturally, if two stack contexts $\langle \kappa, b \rangle$ and $\langle \kappa', b \rangle$ achieve the same expected total reward from each exit of the block $b$, then both the optimal strategy and the expected total reward, are the same.

This simple but precise and effective abstraction of stacks with exit values allows us to consider the following optimality equations $\text{OPT}_{\text{cont}}(M)$. For every $1 \le i \le k$, $q \in Q_i$, $\mathbf{v} \in \mathbb{R}^{|Ex_i|}$:

$$
x(\mathbf{v}, q) = \begin{cases}
x(\mathbf{v}', en)[\mathbf{v}' \mapsto (x(\mathbf{v}, q'))_{q' \in \text{Ret}_b}] & q = (b, en) \in \text{Call} \\
\mathbf{v}(q) & q \in Ex \\
\max_{a \in A(q)} \left\{ r(q, a) + \sum_{q' \in Q} p(q'|q, a) x(\mathbf{v}, q') \right\} & \text{otherwise.}
\end{cases}
$$

Here $\mathbf{v}$ is a vector where $\mathbf{v}(ex)$ is the (expected) reward that we get once we reach exit $ex$ of the current component. Informally when a box is called, this vector is being updated with the current estimates of the reward that we get once the box is exited. The $ex$ entry of this vector $\mathbf{v}' = (x(\mathbf{v}, q'))_{q' \in \text{Ret}_b}$ is $x(\mathbf{v}, (b, ex))$, which is the value that we achieve from exit $(b, ex)$.

This continuous-space abstraction of the countably infinite space of the stack contexts enables the application of deep feedforward neural networks [18] with a finite state encoding in RL. It also provides an elegant connection to the smoothness of differences to exit values: if all exit costs are changed by less than $\varepsilon$, then the cost of each state within a box changes by less than $\varepsilon$, too. The following theorem connects both versions of optimality equations.

**Theorem 3** (Fixed Point). *If $y$ is a fixed point of $OPT_{\text{recur}}$ and $x$ is a fixed point of $OPT_{\text{cont}}$, then $y(\langle \emptyset \rangle, q) = x(\mathbf{0}, q)$. Moreover, any policy optimal from $(\mathbf{0}, q)$ is also optimal from $(\langle \emptyset \rangle, q)$.*

We design a generalization of the Q-learning algorithm [40] for recursive MDPs based on the optimality equations $\text{OPT}_{\text{cont}}$ shown in Algorithm 1. We implement several optimizations in our algorithm. We assume implicit transitions from the entry and exit ports of the box to the corresponding entry and exit nodes of the components. A further optimization is achieved by applying a dimension reduction on the representation of the exit value vector $\mathbf{v}$ by normalizing these values in such a way that one of the exits has value $0$. This normalization does not affect optimal strategies as, when two stacks incur similar costs in that they have the same offset between the cost of each exit, the optimal strategy is still the same, with the difference in cost being this offset.

While the convergence of Algorithm 1 is not guaranteed for the general multi-exit RMDPs, the algorithm converges for the special cases of deterministic proper RMDPs and 1-exit RMDPs (Section 4). For the deterministic multi-exit case, the observation is straightforward as the properness assumption reduces the semantics to be directed acyclic graph, and the correct values are eventually propagated from the leafs backwards.

**Theorem 4.** *Tabular Recursive Q-learning converges to the optimal values for deterministic proper multi-exit RMDPs with a learning rate of $1$ when all state-action pairs are visited infinitely often.*

**Algorithm 1:** Recursive Q-learning

---

1 Initialize $Q(s, v, a)$ arbitrarily
2 **while** *not converged* **do**
3     $v \leftarrow \mathbf{0}$
4     $\text{stack} \leftarrow \emptyset$
5     Sample trajectory $\tau \sim \{(s, a, r, s'), ...\}$
6     **for** $s, a, r, s'$ *in* $\tau$ **do**
7         Update $\alpha_i$ according to learning rate schedule
8         **if** *entered box* **then**
9             $\{s_{\text{exit}_1}, \ldots, s_{\text{exit}_n}\} \leftarrow \text{getExits}(s')$
10            $v' \leftarrow [\max_{a' \in A(s_{\text{exit}_1})} Q(s_{\text{exit}_1}, v, a'), \ldots, \max_{a' \in A(s_{\text{exit}_n})} Q(s_{\text{exit}_n}, v, a')]$
11            $v'_{\min} \leftarrow \min(v')$
12            $v' \leftarrow v' - v'_{\min}$
13            $Q(s, v, a) \leftarrow (1 - \alpha_i)Q(s, v, a) + \alpha_i(r + \max_{a' \in A(s')} Q(s', v', a') + v'_{\min})$
14            $\text{stack.push}(v)$
15            $v \leftarrow v'$
16         **else if** *exited box* **then**
17            $\{s_{\text{exit}_1}, \ldots, s_{\text{exit}_n}\} \leftarrow \text{getExits}(s)$
18            Set $k$ such that $s' = s_{\text{exit}_k}$
19            $Q(s, v, a) \leftarrow (1 - \alpha_i)Q(s, v, a) + \alpha_i(r + v(k))$
20            $v \leftarrow \text{stack.pop}()$
21         **else**
22            $Q(s, v, a) \leftarrow (1 - \alpha_i)Q(s, v, a) + \alpha_i(r + \max_{a \in A(s')} Q(s', v, a'))$
23         **end**
24     **end**
25 **end**
26 **return** $Q$

---

## 4   Convergence of Recursive Q-Learning for Proper 1-exit RMDPs

Recall that a proper 1-exit RMDP is a proper RMDP where, for each component $M_i$, the set of exits $Ex_i$ is a singleton. For this special case, we show that the recursive Q-learning algorithm converges to the optimal strategy. The optimality equations $\text{OPT}_{\text{cont}}(M)$ (similar to [15]) can be simplified in the case of 1-exit RMDPs whose unique fixed point solution will give us the optimal values of the total reward objective. For every $q \in Q$:

$$
x(q) \quad = \quad \begin{cases} x(en) + x(b, ex') & q = (b, en) \in \text{Call}, ex = Ex_{Y(b)} \\ \max_{a \in A(q)} \Big\{ r(q, a) + \sum_{q' \in Q} p(q'|q, a)x(q') \Big\} & \text{otherwise.} \end{cases}
$$

We now denote the system of all these equations in a vector form as $\bar{x} = F(\bar{x})$. Given a 1-exit RMDP, we can easily construct its associated equation system above in linear time.

**Theorem 5** (Unique Fixed Point). *The vector consisting of* $\text{ETotal}^M(q)$ *values is the unique fixed point of $F$. Moreover, a solution of these equations provide optimal stackless strategies.*

Note that for the 1-exit setting, Algorithm 1 simplifies to Algorithm 2 since $v$ is always 0 and $v_{\min}$ is always the maximum Q-value for the exit. The convergence of the recursive Q-learning algorithm for 1-exit RMDPs follows from Theorem 5 and stochastic approximation [40, 8].

**Theorem 6.** *Algorithm 2 converges to the optimal values in 1-exit RMDP when the learning rates satisfy $\sum_{i=0}^{\infty} \alpha_i = \infty$, $\sum_{i=0}^{\infty} \alpha_i^2 < \infty$, and all state-action pairs are visited infinitely often.*

In order to show efficient PAC learnability for $\epsilon$-proper 1-exit RMDP $M$, it suffices to know an upper bound on the expected number of steps taken by $M$ when starting at any vertex with the empty stack content, which will be denoted by $K$.

**Theorem 7** (Efficient PAC Learning for 1-Exit RMDPs). *For every $\epsilon$-proper 1-exit RMDP with diameter $r_{max}$ and the expected time to terminate $\leq K$, $\text{ETotal}^{\mathcal{M}}(s)$ is efficiently PAC-learnable.*

**Algorithm 2:** Recursive Q-learning (1-exit special case)

1 Initialize $Q(s,a)$ arbitrarily
2 **while** *not converged* **do**
3      Sample trajectory $\tau \sim \{(s,a,r,s'),...\}$
4      **for** $s,a,r,s'$ *in* $\tau$ **do**
5          Update $\alpha_i$ according to learning rate schedule
6          **if** *entered box* **then**
7              $s_{\text{exit}} \leftarrow \text{getExit}(s')$
8              $Q(s,a) \leftarrow$
             $(1-\alpha_i)Q(s,a) + \alpha_i(r + \max_{a' \in A(s')} Q(s',a') + \max_{a' \in A(s_{\text{exit}})} Q(s_{\text{exit}},a'))$
9          **else if** *exited box* **then**
10              $Q(s,a) \leftarrow (1-\alpha_i)Q(s,a) + \alpha_i(r)$
11          **else**
12              $Q(s,a) \leftarrow (1-\alpha_i)Q(s,a) + \alpha_i(r + \max_{a \in A(s')} Q(s',a'))$
13          **end**
14      **end**
15 **end**
16 **return** $Q$

## 5 Experiments

We implemented Algorithm 1 in tabular form as well as with a neural network. For the tabular implementation, we quantized the vector $v$ directly after its computation to ensure that the Q-table remains small and discrete. For the neural network implementation we used the techniques used in DQN [30], replay buffers and target networks, for additional stability. The details of this implementation can be found in the appendix.

We consider three examples: one to demonstrate the application of Recursive Q-learning for synthesizing probabilistic programs, one to demonstrate convergence in the single-exit setting, and one to demonstrate the use of a context-free reward machine. We compare Recursive Q-learning to Q-learning where the RMDP is treated as an MDP by the agent, i.e., the agent treats stack calls and returns as if they were normal MDP transitions.

### 5.1 Cloud computing

The cloud computing example, introduced in Example 1, is a recursive probabilistic program with decision points for an RL agent to optimize over. The optimal strategy is to select the reliable server and to never upgrade. This strategy produces an expected total reward of $-5.3425$. Figure 3 shows that tabular Recursive Q-learning with discretization quickly converges to the optimal solution on this multi-exit RMDP while Q-learning oscillates around a suboptimal policy.

### 5.2 Infinite spelunking

Consider a single-exit RMDP gridworld with two box types, 1 and 2, shown at the bottom of the figure to the right. These box types are the two types of levels in an infinitely deep cave. When falling or descending to another level, the level type switches. Passing over a trap, shown in red, results in the agent teleporting to a random position and falling with probability 0.5.

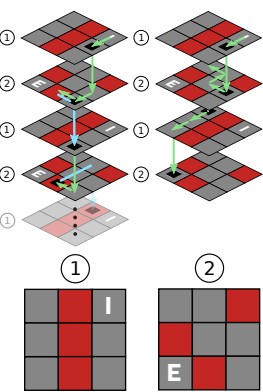

The agent has fallen into the cave at the position denoted by $I$ without climbing equipment. However, there is climbing equipment in one of the types of levels at a known location denoted by $E$. The agent has four move directions—north, east, south, west—as well as an additional action to descend further or ascend. Until the climbing equipment is obtained, the agent can only descend. Once the climbing equipment is obtained, the traps no longer affect the agent and the agent can ascend

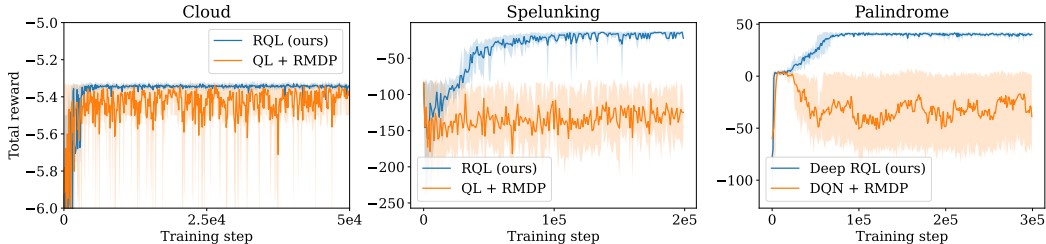

Figure 3: Learning curves. Ten runs were performed for each method. The mean is shown as a dark line. The region between the 10th and 90th percentile is shaded. RQL refers to tabular Recursive Q-learning, QL + RMDP refers to using tabular Q-learning on the RMDP, Deep RQL refers to Deep Recursive Q-learning, and DQN + RMDP refers to using Deep Q-networks on the RMDP.

only from the position where it fell down. With probability $0.01$ the agent ascends from the current level with the climbing gear. This has the effect of box-wise discounting with discount factor $0.99$. The agent's objective is to leave the cave from where it fell in as as soon as possible. The reward is $-1$ on every step.

There are two main strategies to consider. The first strategy tries to obtain the climbing gear by going over the traps. This strategy leads to an unbounded number of possible levels since the traps may repeatedly trigger. The second strategy avoids the traps entirely. The figure to the right shows partial descending trajectories from these strategies, with the actions shown in green, the trap teleportations shown in blue, and the locations the agent fell down from are shown as small black squares. Which strategy is better depends on the probability of the traps triggering. With a trap probability of $0.5$, the optimal strategy is to try and reach the climbing equipment by going over the traps. Figure 1 shows the convergence of tabular Recursive Q-learning for 1-exit RMDPs to this optimal strategy while the strategy learned by Q-learning does not improve.

### 5.3  Palindrome gridworld

To demonstrate the ability to incorporate context-free objectives, consider a $3 \times 3$ gridworld with a goal cell in the center and a randomly selected initial state. The agent has four move actions—north, east, south, west—and a special control action. The objective of the agent is to reach the goal cell while forming an action sequence that is a palindrome of even length. What makes this possible is that when the agent performs an action that pushes against a wall of the gridworld, no movement occurs. To monitor the progress of the property, we compose this MDP with a nondeterministic pushdown automaton. The agent must use its special action to determine when to resolve the nondeterminism in the pushdown automaton. Additionally, the agent uses its special action to declare the end of its action sequence. To ensure properness, the agent's selected action is corrupted into the special action with probability $0.01$. The agent is given a reward of $50$ upon success, $-5$ when the agent selects an action that causes the pushdown automaton to enter a rejecting sink, and $-1$ on all other timesteps.

Figure 3 shows the convergence of Deep Recursive Q-learning to an optimal strategy on this example, while DQN fails to find a good strategy.

## 6  Related Work

Hierarchical RL is an approach for learning policies on MDPs that introduces a hierarchy in policy space. There are three prevalent approaches to specify this hierarchy. The options framework [37] represents these hierarchies as policies each with a starting and termination condition. The hierarchy of abstract machines (HAM) framework [31] represents the policy as as a hierarchical composition of nondeterministic finite-state machines. Finally, the MAXQ framework [13] represent the hierarchy using a programmatic representation with finite range variables and strict hierarchy among modules. Semi-Markov decision processes (SDMPs) and hierarchical MDPs are fundamental models that appear in the context of hierarchical RL. SMDPs generalize MDPs with timed actions. The RL algorithms for SMDPs are based on natural generalization of the Bellman equations to accommodate

timed actions. Hierarchical MDPs model bounded recursion and can be solved by flattening to a MDP, or by producing policies that are only optimal locally.

Recursive MDPs model *unbounded* recursion in the environment space. The orthogonality of recursion in environment space and in policy space means they are complementary—one can consider applying ideas in hierarchical RL to find a policy in an RMDP. The authors of [3] proposed using partially specified programs with recursion to constrain the policy space, but only considered bounded recursion. Hierarchical MDPs [37, 31, 13], and factored MDPs [12, 20] indeed offer compact representations of finite MDPs. These representations can be exponentially more succinct. However, note that finite instances of these formalism are not any more expressive than finite MDPs as instances of these formalism can always be rewritten as a finite MDP. On the other hand, recursive MDPs studied in this paper are strictly more expressive than finite MDPs. Even 1-exit RMDPs may not be expressible as finite MDPs, due to a potentially unbounded stack, but remarkably they can be solved exactly with a finite tabular model-free reinforcement learning algorithm (Theorem 6) without needing to resort to ad-hoc approximations of the unbounded stack configurations.

Context-free grammars in RL for optimization of molecules has been considered before by introducing a bound on the recursion depth to induce a finite MDP [25, 42]. Combining context-free grammars and reward machines was proposed as a future research direction in [24], where context-free reward machines have been first described in this paper. Recursive MDPs have been studied outside the RL setting [16], including results for 1-exit RMDP termination [16] and total reward optimization under positive reward restriction [15]. A convergent model-free RL algorithm for 1-exit RMDPs with positive cost restriction was proposed in [21]. The results on undecidability, on PAC-learnability, the introduction of the algorithm Recursive Q-learning, convergence results for Recursive Q-learning, and its deep learning extension are novel contributions of this paper.

## 7 Conclusion

Reinforcement learning so far has primarily considered Markov decision processes (MDPs). Although extremely expressive, this formalism may require "flattening" a more expressive representation that contains recursion. In this paper we examine the use of recursive MDPs (RMDPs) in reinforcement learning—a setting we call recursive reinforcement learning. A recursive MDP is a collection of MDP components, where each component has the ability to recursive call each other. This allows the introduction of an unbounded stack. We propose abstracting this discrete stack with a continuous abstraction in the form of the costs of the exits of a component. Using this abstraction, we introduce Recursive Q-learning—a model-free reinforcement learning algorithm for RMDPs. We prove that tabular Recursive Q-learning converges to an optimal solution on finite 1-exit RMDPs, even though the underlying MDP has an infinite state space. We demonstrate the potential of our approach on a set of examples that includes probabilistic program synthesis, a single-exit RMDP, and an MDP composed with a context-free property.

**Acknowledgments.** ⬚ This project has received funding from the European Union's Horizon 2020 research and innovation programme under grant agreements 864075 (CAESAR), and 956123 (FOCETA). This work is supported in part by the National Science Foundation (NSF) grant CCF-2009022 and by NSF CAREER award CCF-2146563. This work utilized the Summit supercomputer, which is supported by the National Science Foundation (awards ACI-1532235 and ACI-1532236), the University of Colorado Boulder, and Colorado State University. The Summit supercomputer is a joint effort of the University of Colorado Boulder and Colorado State University.

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
