# OpenReview forum: "Recursive Reinforcement Learning"
_NeurIPS.cc/2022/Conference — NeurIPS 2022 Accept_

### Official Review · Reviewer_pptt · 2022-07-01

**Rating:** 3
**Confidence:** 3
**Soundness:** 3 good
**Presentation:** 3 good
**Contribution:** 2 fair

**Summary:**

The main idea of this paper is to formulate Q-learning on a type of MDP with a recursive structure, akin to the "call stack" of operating systems. The two arguments are that (1) many realistic MDPs have this sort of structure, and that (2) an RL algorithm designed to exploit this structure will perform better than algorithms with such knowledge. The paper includes some theoretical results about when such recursive MDPs are learnable (similar to standard computability results) and then proposes a variant of Q-learning for solving these recursive MDPs. The variant of Q-learning looks like standard Q-learning, with extra bookkeepping to keep track of the stack trace. On three didactic problems, the proposed method outperforms RL methods that do not exploit the recursive problem nature.

**Questions:**

Some questions that would be good to address:
1. Is it easier or harder for users to define a recursive MDP than a flat MDP?
2. What, precisely, is the relationship with hierarchical RL? Is the recursive MDP a special case of hierarchical RL where the structure is manually provided?
3. If the recursion depth is limited (say, to 3 levels), how beneficial is the proposed framework over standard Q-learning? Note that today's computers seem to work just fine with limited recursion depth.

**Limitations:**

I did not see any explicit sections on limitations.

**Strengths And Weaknesses:**

Strengths
* The paper is generally well written, with compelling motivation for why some problems should be treated as recursive MDPs.
* The potential applications discussed on page 3 are great for explaining the motivation for the recursive MDP.
* The experiments in Fig. 3 nicely illustrate three applications.
* The paper is novel to the best of my knowledge. Formalizing the connections between MDPs and computability seem potentially impactful.

Weaknesses
* The tradeoffs involved in the paper are a bit unclear. See questions below.
* The proposed method was a bit hard to follow. Are there some special cases where the proposed method is equivalent to Q-learning? (e.g., if the observation is augmented with the stack trace)?
* The formal definition of recursive MDPs was complicated and hard to follow (esp. L124 -- L135)

Minor comments
* L1 -- I didn't understand this sentence
* L3 "they must rely on the practitioner's ingenuity" -- Nicely put!
* L23 "regularity assumptions" -- Cite.
* L32 "to generalize its learning upon" -- I didn't understand this sentence
* L41 "Unlike ... enables generalizability" -- Where is this demonstrated/proven?
* L83 "single-exit" -- Be consistent with "1-exit" and "single-exit"
* Fig 1, Fig 2 -- I found these figures pretty hard to parse.
* L94 -- It wasn't clear to me why the charging example requires recursion. Shouldn't it be possible to formulate this as a simple, flat MDP?
* L127 -- L129 -- I found this discussion confusing.
* L318 -- If this task requires memory, then using a Markovian Q-learning policy as a baseline seems a bit unfair.

---

> ### Author Response · Authors · 2022-08-02
> **Reply to Reviewer pptt**
>
> Thank you for your careful review and helpful comments. We will implement your feedback in the final version of this paper.
>
> *Are there some special cases where the proposed method is equivalent to Q-learning?*
>
> Yes, when applied to an RMDP with no calls, i.e. an MDP, Recursive Q-learning reduces to standard Q-learning, where the discount factor has been set to $1$ and the state has been augmented with the constant exit value of $0$. We can set the discount factor to $1$ because of the properness assumption about the policies, e.g. discounting has been baked into the model as probabilistic termination.
>
> *L94 -- It wasn't clear to me why the charging example requires recursion. Shouldn't it be possible to formulate this as a simple, flat MDP?*
>
> In this case, we are not assuming any upper bound on the capacity of the battery, which induces an unbounded stack.
>
> *L318 -- If this task requires memory, then using a Markovian Q-learning policy as a baseline seems a bit unfair.*
>
> In our initial experiments, we found that augmenting the state space of the Q-learner with the unbounded stack resulted in worse performance early in learning, compared with applying the Q-learning update rule to the RMDP structure. We would be happy to include these results.
>
> Answers to questions:
>
> 1. A flat, finite-state MDP can simply be written down as an RMDP with a single component and no calls, so it is not any harder. Critically, RMDPs allow us to describe infinite objects finitely. There are other mechanisms for defining infinite objects finitely, like augmenting finite MDPs with unbounded domain variables — and these may occasionally be more convenient than recursion, but like with programming languages, recursion often proves convenient.
>
> 2. Hierarchical RL is concerned with using a bounded hierarchy in policy space to more efficiently solve (finite) MDPs. A fundamental object associated with hierarchical RL, a finite hierarchical MDP, is no more expressive than finite MDPs. In fact, hierarchical MDPs are a special case of recursive MDPs. In this paper, we demonstrate that learning globally optimal policies on certain infinite MDPs, in the form of 1-exit RMDPs, can be done exactly.
>
> 3. While the maximum recursion depth is always finite in practice, it is big enough to make such an approach infeasible. For instance, already for an 1-exit RMDP with two boxes and a stack depth of 20 the unfolded MDP would have at least one million times more states than the original RMDP we started with. Our Recursive Q-learning is not much more complex than the standard Q-learning algorithm, so we expect our approach to be thousands of times faster. Besides, one does not typically look for a complex optimal policy that just works for one stack depth, but not for another. Our solution not only deals with arbitrary stack depths, but also finds simple (stackless and memoryless) optimal policies for 1-exit RMDPs.

---

> > ### Comment · Reviewer_pptt · 2022-08-05
> > **Reviewer response**
> >
> > Thank you for the detailed answers to the questions!
> >
> > Broadly, I remain a bit unsure about the usefulness of recursive RL. Part of the concern stems from the infinite recursion -- I'm unsure how practically useful this is. Part of the concern is that the proposed method seems to require privileged information about the _structure_ of the MDP, an assumption that standard RL algorithms don't seem to require. It would be great if the authors could discuss address these concerns. Providing some practical examples might be helpful.

---

> > > ### Author Response · Authors · 2022-08-08
> > > **Reply to Reviewer pptt**
> > >
> > > Thank you for your questions.
> > >
> > > Concerning the usefulness of unbounded recursion:
> > > By allowing unbounded (infinite) recursion we gain a few benefits. First, we don’t require a bound on the stack to be provided as a hyperparameter that may potentially need to be tuned. Second, the learning agent is always performing in its domain, i.e., we don’t need to specifically handle the case where the stack grows beyond the specified bound. And most importantly, our learning algorithm is simpler and can scale better compared to a learning algorithm which requires a bound and then treats the problem as a standard MDP. This latter algorithm provides a solution where the state space grows exponentially in the bound provided for the stack — a curse of dimensionality that we show can be completely avoided in the 1-exit setting. The resulting policies are simpler (stackless) and more general (the stack can be unbounded). It’s worth noting that under our assumptions, all executions take a finite amount of time with probability 1. However, this does not imply executions are bounded.
> > >
> > > Concerning privileged information:
> > > Our proposed method does not require knowing specific internal structures and probabilities. It only requires knowing the type of the transition (“call”, “return”, and “normal”) at the time it occurs. The “call” transitions need to be augmented with information about the possible return values of the component MDP that was called. If desired, one can decide to learn the nature of the exits through experience by storing the exits as they are discovered.
> > >
> > > When formulating a real problem as an MDP, there is always a modeler sitting in the loop. For example, on a robotics task, a modeler must decide what the observables are (which should includes velocities of the robot if one wants the problem to be Markovian), how much time should elapse between timesteps (MDPs are discrete time while the robot is operating in continuous time), and what the rewards should be. The power of RL is that the learning algorithm can then automate the design of the controller once the problem is formulated. With RMDPs, we add another tool found in programming to the modeler’s toolbox: unbounded recursion. Recursion comes from self-similarity in the system, which may come from imposing programmatic structures or appear naturally in the system.
> > >
> > > One case in which recursion appears naturally is in examining reproducing biological populations. Let’s discuss a concrete example.
> > >
> > > Consider that we have a population of two kinds of organisms: green (G) and blue (B). Under one environment condition (say, “a”) the green organisms can split into two green organisms with some unknown probability, or transform into a blue organism with the remaining probability. We can write this behavior like so: G -(a)-> p: GG + (1-p): B. All rules for this population, which are unknown to the agent and to the modeler, may look like the following:
> > >
> > > G -(a)-> p: GG + (1-p): B
> > >
> > > G -(b)-> 1: GB
> > >
> > > B-(a)-> epsilon (dies without reproducing)
> > >
> > > B-(b)-> epsilon (dies without reproducing)
> > >
> > > Assume that there is some reward associated with each transition, and starting from a population consisting of a single green organism, the objective of the RL agent is to find a policy to maximize the sum of the rewards.
> > >
> > > Notice that this system can be modeled as an RMDP with 1-exit, where we have one box per organism. When the RL agent observes the production of more organisms as the result of an action, this transition looks like a “call” to one of the organisms produced with the other organisms called later in sequence. For instance, observing the production of two green organisms after executing action “a” can be modeled as calling two “G” boxes in sequence before going out the unique exit of the module. The RL agent will learn from experience which scenarios occur and their probabilities.
> > >
> > > While performing RL, the agent only needs to know the different types of organisms in the system and not their transition structure. That will be observed when the agent explores the system. For instance, starting from a single organism and choosing action “a”, the RL agent may observe the transformation into a certain population. The RL algorithm need not know the connectivity in advance, but only an ability to distinguish different organisms produced in a transition.
> > >
> > > Note that this example can’t be modeled as a finite-state MDP, as the number of organisms present in the system may be unbounded, and each such organism is a finite-state MDP in itself. The modeler imparts the prior knowledge that all blue/green organisms are identical in behavior to all other blue/green organisms. In this example, the modeler expresses this self-similarity with recursion.

---

> > > > ### Comment · Reviewer_pptt · 2022-08-09
> > > > **Reviewer response**
> > > >
> > > > Thanks for continuing the discussion!
> > > >
> > > > > Concerning the usefulness of unbounded recursion
> > > >
> > > > One clarification question about this: in either the paper or one of these discussion threads, it was mentioned that the reward function doesn't depend on the stack. If I recall correctly, one baseline used in the paper was a standard RL agent with observations augmented to include the stack trace. Here's a similar baseline (apologies if this was already mentioned): a standard RL agent with only local information (i.e., no information about the stack trace). Are there scenarios where this agent would fail? [No need to provide a detailed description. Yes/No with a 1-sentence intuition is fine]
> > > >
> > > > > When formulating a real problem as an MDP, there is always a modeler sitting in the loop.
> > > >
> > > > I found this argument very compelling!
> > > >
> > > > > Consider that we have a population of two kinds of organisms
> > > >
> > > > Would it be possible to provide a more realistic example? E.g., some application where _humans_ have tried applying RL in the past, or would like to?

---

> > > > > ### Author Response · Authors · 2022-08-09
> > > > > **Reply to Reviewer pptt**
> > > > >
> > > > > Yes, this would fail even in the 1-exit case due to the need to handle calling components specially.
> > > > >
> > > > > In more detail: In the 1-exit setting, once one makes the observation that the stack information is not required, one may consider solving each individual component MDP separately, where the exit is treated as a terminal state. However, treating these as MDPs is not enough due to the need to deal with calling other components. Note that the expected reward-to-go when calling another component consists of two parts: 1) the expected reward accumulated in the called component (this is the Q-value of the entry point of the called component) plus 2) the expected reward accumulated to the exit of the current component after the called component has returned (the Q-value from the exit of the called component). Adding this to the Q-learning update rule leads directly to Algorithm 2.
> > > > >
> > > > > Here are some of the examples where various authors have expressed a desire for a formalism that is similar to recursive RL:
> > > > >
> > > > > **Reward Machines: Exploiting Reward Function Structure in Reinforcement Learning** — Icarte et al. (https://www.jair.org/index.php/jair/article/view/12440/26759)
> > > > >
> > > > > > “Finally, we see many opportunities for using formal languages to define correct reward specification via reward machines and defining novel RL methodologies to exploit the knowledge within reward machines – resulting in agents that can understand humans’ instructions and use them to solve problems faster.  To keep pushing in this direction, it might be worth exploring the potential benefits of ascending the Chomsky hierarchy and studying combinations of reward machines with context-free and context-sensitive grammars"
> > > > >
> > > > > We describe context-free reward machines — reward machines with a pushdown stack — as an application of RMDPs in the introduction of our paper.
> > > > >
> > > > > **Grammars and reinforcement learning for molecule optimization** — Kraev (https://arxiv.org/pdf/1811.11222.pdf)
> > > > >
> > > > > > “We consider that a context free grammar (CFG) is a very natural way of representing an acyclic graph. We formulate a new grammar for SMILES strings that guarantees correct atom valences by construction, and allows for arbitrarily long, branching chains of aliphatic cycles, and the more common kinds of aromatic cycle structures. Due to the recursive nature of a CFG, we can represent a combinatorially large number of possible cycle structures with a small number of expansion rules (for example, an aliphatic 5-cycle can consist of any 5 sub-structures with 2 free valences each, some of which can also form part of other cycles)”
> > > > >
> > > > > > “... we decide to try a different tack, namely consider the molecule generation problem as a reinforcement learning problem, and optimize using best-of-batch policy gradient. ..”
> > > > >
> > > > > > “To address the issues identified by [Kusner et al., 2017] we introduce additional masking into the process of selecting the next production rule. One kind of masking tracks the ‘terminal distance’ of the current sequence, making sure the rule expansion terminates before the maximum number of steps allowed by the model.”
> > > > >
> > > > > In this paper authors explore designing a molecule using reinforcement learning where the allowed choices (environment) is expressed via a context-free grammar. The solution approach considers introducing a bound (an upper bound on the number of steps) to recover essentially a finite-state MDP.
> > > > >
> > > > > A similar idea is explored by Zhou et al. in their Nature article **Optimization of Molecules via Deep Reinforcement Learning** (https://www.nature.com/articles/s41598-019-47148-x.pdf)
> > > > >
> > > > > A similar application is studied by Wu et al. in **REINAM: Reinforcement Learning for Input-Grammar Inference** (https://dl.acm.org/doi/pdf/10.1145/3338906.3338958) where RL is used to infer grammar (recursive MDPs can faithfully encode CFGs under stochastic productions).
> > > > >
> > > > >
> > > > > Another instance comes from Dietterich’s **Hierarchical Reinforcement Learning with the MAXQ Value Function Decomposition** (https://www.jair.org/index.php/jair/article/view/10266/24463 ) where on p. 238 the author explicitly disallows recursion.
> > > > >
> > > > > > “Stated another way no subtask can invoke itself recursively either directly or indirectly”.
> > > > >
> > > > > RMDPs free the practitioner from requiring this assumption.

---

### Official Review · Reviewer_wbL5 · 2022-07-02

**Rating:** 6
**Confidence:** 1
**Soundness:** 4 excellent
**Presentation:** 4 excellent
**Contribution:** 3 good

**Summary:**

This paper presents an analysis of a special class of infinite MDPs, recursive MDPs. Recursive MDPs are represented as a collection of components, which have nodes and boxes. Boxes call other components and have associated "call ports" and "return ports". Actions in each component affect transitions to nodes and boxes. A component reward function is defined over component transitions. The components then implicitly define an infinite MDP whose (countable) state space is defined by pairs of box stacks and nodes. Importantly, the stack acts as a context analogous to a stack in a pushdown automaton and can grow infinitely (this is the sense in which recursion is implemented).

The authors analyze RMDPs under two assumptions: 1 - all RMDP strategies have finite return (this can be enforced with a discount) and 2 - RMDPs are epsilon-proper and all policies have a fallign expected stack height. They then report a recursive Q-learning algorithm for multi-exit RMDPs based on an abstraction of the call stack. Specifically, call-stacks are equivalent if they share an exit-value--that is, the vector of total expected rewards from the exit nodes of a box given the rest of the stack. Theorem 3 states that the fixed point of the abstract/continuous version of the RMDP is equivalent to that of the original RMDP, which makes it possible to apply algorithms that operate over a fixed-size state representation (eg standard deep RL methods). Their algorithm (alg 1) is not guaranteed to converge for general multi-exit RMDPs, but they report a version (alg 2) that converges for single-exit RMDPs (single-exit RMDPs only have components with a single exit node, are more expressive than finite-MDPs, and are related to "controlled branching processes"). Experiments are then reported on 3 domains, including one in which a DQN-based learning algorithm is used. Compared to using standard Q-learning/DQN, their recusive Q-learning algorithm does better.



**Questions:**

I only have a few clarifying questions:
- Were the domains 5.1 and 5.3 general RMDPs? It would be helpful if this were stated explicitly.
- Stackless strategies are mentioned on lines 166 and then in Theorm 5 in relation to recursive Q-learing applied to 1-exit RMDPs. Do the other results apply to general stack-based strategies?

**Limitations:**

The authors have been up front about where their method applies and its limitations.

**Strengths And Weaknesses:**

Originality
- The paper provides an analysis of a novel and interesting class of MDPs

Quality
- The types of analyses reported are not in my main field of study, so I cannot comment on whether they meet technical standards typical for this subfield, but I found them to be a sensible way to approach the problem

Clarity
- Given the technical nature of the paper, I thought the main ideas and results were presented clearly

Significance
- The authors note in the introduction that recursive RL can be applied to probabilistic program synthesis, context-free reward machines, and stochastic context-free grammars. These are important domains, but the main test cases they apply their method were mainly toy domains.

---

> ### Author Response · Authors · 2022-08-02
> **Reply to Reviewer wbL5**
>
> Thank you for your careful review and helpful comments. We will implement your feedback in the final version of this paper.
>
> Answers to questions:
>
> 1. Yes, 5.1 and 5.3 were multi-exit RMDPs while 5.2 was a 1-exit RMDP. We will state this more clearly in the final version.
>
> 2. We have optimal stackless strategies only in the case of 1-exit RMDPs. Unless stated otherwise, we consider general strategies. We show in the appendix that for 1-exit RMDPs (under Assumption 1) there always exists an optimal strategy that is stackless and we are able to reconstruct it easily from the fixed point solution of F (Theorem 5). The optimal strategies in deterministic RMDPs (Theorem 4) may require finite memory. The undecidability of the strategy existence problem (Theorem 1) holds for general stack-based strategies.

---

> > ### Comment · Reviewer_wbL5 · 2022-08-09
> > **Reply**
> >
> > Thanks to the authors for clarifying these points!

---

### Official Review · Reviewer_aWu3 · 2022-07-10

**Rating:** 4
**Confidence:** 4
**Soundness:** 3 good
**Presentation:** 3 good
**Contribution:** 2 fair

**Summary:**

This paper studies reinforcement learning in recursive Markov decision processes (RMDPs). RMDPs were proposed a few years ago, and the authors cited one paper in which reinforcement learning for a special type of RMDPs (1-exit RMDPs with a positive reward restriction) was presented. The goal of the current paper is to study reinforcement learning in more general RMDPs. The paper offers both theoretical and empirical contributions. A number of theoretical assertions are made, e.g., about undecidability or PAC-learnability. The RMDPs can model infinite state spaces, but the algorithms for solving them have to deal with infinite quantities in one way or the other. For this reason, an approximate algorithm for RMDPs is proposed in this paper. Empirical results are presented and the new method is compared with standard flat Q-learning.

**Questions:**

Line 36 criticises HRL for not being able to model infinite recursion? But, the new method has the same problem since the stack can be infinite too? Is this claim justified then?

Example 1 is great. Have you tried to find a hierarchical or maybe a PDDL-like method in RL that could model recursion? Having an integer variable seems to be sufficient.

I am not sure why HRL or factored RL methods cannot cope with the example presented in lines 94 to 108. Could you please clarify?

Why the new definition of proper strategies is required? In standard MDPs (see [26]), discounting is used to cope with summations over infinite sets. The authors talk about the discount factor in the appendix, but the definition of the proper polices seems to be redundant. I believe that the entire theory could be presented without this concept. It adds a lot of confusion.

Why the comparisons are only against flat Q-learning?

I am not sure why Q-learning oscillates in section 5.1? Given a sufficient amount of exploration, Q-learning is guaranteed to find an optimal policy. If Q-learning is put in a disadvantaged position here, the authors should implement Q-learning in a way that will allow it to use the same information as their method based on RMDPs.

It would be useful if the paper was more explicit about which concepts are known and which were proposed in this paper.

**Limitations:**

n.a.

**Strengths And Weaknesses:**

This paper offers a competent discussion of RMDPs and adds useful extensions to reinforcement learning algorithms for solving these models. If we disregard the broader reinforcement learning literature, this paper looks like a strong contribution. This work on RL in RMDPs is, however, not placed correctly in the context of the existing methods in the reinforcement learning community. The biggest issue is that the current narrative of criticism of flat reinforcement learning muddles the waters, which makes it impossible to see how this work can be placed in the context of the existing literature. In several parts of the paper, the authors refer to flat RL saying that flat RL cannot cope with certain properties of the tasks that can be addressed by RMDPs. The problem is that there exist powerful methods in the RL literature that can cope with structured domains. Hierarchical reinforcement learning or representations with state features can exploit the fact that the state space is composed of a number of state features. This makes the empirical results in this paper unfair since the comparisons should be against hierarchical or factored algorithms. Comparisons with flat Q-learning are not sufficient since there exist algorithms that are at least as intelligent as the methods proposed here. This is certainly authors responsibility to find out how exactly their new method relates to the existing methods.

RMDP is an interesting model, but solving it has its own challenges. For example, to solve the example presented in Fig. 2, it would be possible to use a  RL algorithm with state features, and to use an integer variable to count how many times S() was executed. This would introduce a possibly infinite number of states in the representation, but the size of the stack has exactly the same problem since S() can be executed infinitely many times. So, the stack will grow too.

The relationship of this method with the existing RL literature is not clear, and it difficult to judge if the method adds anything that is not used in the current reinforcement learning methods.

---

> ### Author Response · Authors · 2022-08-02
> **Reply to Reviewer aWu3 (part 1)**
>
> Thank you for your careful review and helpful comments. We will implement your feedback in the final version of this paper.
>
> Finite hierarchical MDPs and finite factored MDPs offer compact representations of finite MDPs. These representations can be exponentially more succinct. However, note that finite instances of these two formalisms are not strictly more expressive than finite MDPs because instances of these formalisms can always be rewritten as a finite MDP. On the other hand, finite recursive MDPs are strictly more expressive than finite MDPs. A 1-exit finite RMDP may not be able to be expressed as a finite MDP, due to a potentially infinite stack, but nonetheless can still be solved exactly with a finite tabular model-free reinforcement learning algorithm. This is the algorithm we have proposed in this paper. It turns out that the infiniteness of the MDP corresponding to a 1-exit finite RMDPs does not yield intractability in learning. When we refer to the unsuitability of existing methods, it is this distinction between the convergence to the global optimum of our method in 1-exit finite RMDPs versus other techniques, which are forced to resort to approximation in this setting. Note that our results here are distinct from the referenced prior work in BMDPs, which makes a positive cost assumption that we do not make here. For the general multi-exit setting, we have shown that achieving exact results, even with a finite RMDP, is impossible. We have shown two directions to handle this impossibility. First, we have shown that producing $\varepsilon$-optimal policies is possible, which enables a PAC learning approach. Second, we have proposed using a compact vector abstraction of the stack which can be used with neural networks. We will clarify these points in the final version.
>
> Answers to questions:
>
> 1. In the 1-exit setting, we are able to model infinite recursion without sacrificing exact convergence to the global optimum. The lack of need to perform any ad-hoc approximation of this infinite stack with our tabular model-free reinforcement learning method in this setting is what we are referring to when we refer to existing methods as unsuitable.
>
> 2. This paper aims to develop theoretical foundations of reinforcement learning for recursively represented infinite-state MDPs. To the best of our knowledge, there are no convergent RL algorithms for this class in prior work. We develop a theory for this setting and create the first convergent algorithm for finite 1-exit RMDPs with no assumption on its cost.
>
> 3. The extension here is to take a reward machine (a finite state machine with reward) and add an unbounded pushdown stack. This allows us to model pushdown automata that accept context-free languages. When composed with a finite MDP (we synchronize their transitions and concatenate their states), this yields a finite RMDP with an unbounded stack. Note that in the presence of stochasticity, there may be no bound for the stack height once composed with the MDP. Our previous discussion on the applicability of existing techniques then applies here.
>
> 4. There are multiple ways of discounting in RMDPs. The most straightforward way, discounting each timestep that elapses, has a distinct disadvantage — this discounting induces a multi-exit RMDP and policies may depend on the stack. Alternative ways of discounting, like box-wise discounting preserve the number of exits of the RMDP and maintain tractability. By considering proper policies, our results generalize all of these discounting techniques, including ones we have not discussed.
>
> 5. As discussed above, the applicability of existing techniques for the general RMDP problem is unclear. Many hierarchical RL methods for finite MDPs reduce their problem to one of a finite semi-Markov decision process (SMDP), where a variant of Q-learning that allows variable transition time is used. If the transition time is always $1$, as it is in our case, this algorithm reduces to standard Q-learning.
>
> (continued in next reply)

---

> > ### Author Response · Authors · 2022-08-02
> > **Reply to Reviewer aWu3 (part 2)**
> >
> > Answers to questions (continued):
> >
> > 6. As mentioned in lines 280-282, we apply Q-learning to the RMDP transition system where all transitions are treated as normal MDP transitions. This was designed to answer the following: If one replaces the update rules we’ve introduced in Recursive Q-learning (3 separate update rules, the vector $v$ is included in the updates) with just the standard Q-learning update rules, is performance empirically the same? Our experiments show that this is not the case. The reason that Q-learning oscillates in this case is that Q-learning is reasoning over an infinite POMDP, where the observation is the current state and the unobserved component is the unbounded stack. We found in preliminary experiments that appending the entire stack to the state information to form an infinite MDP resulted in worse performance initially with slow improvement over time. We would be happy to include these results. It is worth noting that the representation of the policy grows forever as we expand the Q-table, under this representation. For Recursive Q-learning, our abstraction of the stack as a vector allows us to perform effective discretization of this infinite space. Although Q-learning has been shown to converge for countably infinite MDPs if all state-action pairs are visited infinitely often, results on the finite time behavior of Q-learning have only been shown on finite MDPs.
> >
> > 7. While the model of recursive Markov decision processes was already introduced by Etessami and Yannakakis [Ref 14 in our submission], it was not studied in the context of model-free reinforcement learning. A recent work studies a much restricted model of branching Markov decision processes [14 in our submission] with strictly positive cost assumption. The results on undecidability, on PAC-learnability, the introduction of the algorithm Recursive Q-learning, convergence results for Recursive Q-learning, and its deep learning extension are novel contributions of this paper. We will clarify this in the final version.

---

> > > ### Comment · Reviewer_aWu3 · 2022-08-07
> > > **from response to pptt**
> > >
> > > In their response to the reviewer *pptt*, the authors said:
> > >
> > > `Our Recursive Q-learning is not much more complex than the standard Q-learning algorithm, so we expect our approach to be thousands of times faster. Besides, one does not typically look for a complex optimal policy that just works for one stack depth, but not for another. Our solution not only deals with arbitrary stack depths, but also finds simple (stackless and memoryless) optimal policies for 1-exit RMDPs.`
> > >
> > > However, if the algorithm can deal with arbitrary stack depths, and if the reward depends on the state variables that determine the stack depth, then there is either a need to deal with an exponential number of states or a strong approximation is required. I would probably need to review a revised version of this paper again to see if the authors' explanations can make a compelling story.

---

> > > > ### Author Response · Authors · 2022-08-08
> > > > **Reply to Reviewer aWu3**
> > > >
> > > > Your observation that adding a dependency of the rewards on the stack contents results in an unavoidable growth of the state space is correct. However, in RMDPs the reward does not depend on the stack depth, it only depends on the local state of the component MDP the agent is in. Viewing RMDPs as recursive programs, there are no global variables. The agent can only view variables that are in the local scope. Including a dependency of the reward on the stack depth in RMDPs requires passing additional information about the stack to these local variables, e.g., a counter that is passed through the recursive calls. The growth in the state space depends on the complexity of the dependency on the stack, ranging from a constant factor (for instance, the reward depends on the parity of the stack depth) to unbounded (for instance, the reward is inversely proportional to the stack depth).
> > > >
> > > > Our reply to reviewer pptt was in the context of considering (finite) 1-exit RMDPs. An approach for this setting that simply fixes a bound on the stack and then augments the state information with the stack yields a finite MDP whose state space grows exponentially in the bound on the stack. Our approach, on the other hand, shows that the stack information is unnecessary and we propose an exact learning algorithm that learns an optimal strategy with a simple modification to the standard Q-learning update rule. Our proposed algorithm does not need to explore this exponential state space in this setting, improving learning efficiency. Note that this allows us to deal with arbitrary stack depths efficiently (when the rewards do not depend arbitrarily on the stack contents). The lack of dependence of rewards on the stack contents is natural; none of our examples have this dependence.

---

### Official Review · Reviewer_dojD · 2022-07-12

**Rating:** 7
**Confidence:** 3
**Soundness:** 3 good
**Presentation:** 3 good
**Contribution:** 4 excellent

**Summary:**

This paper introduces recursive MDPs (RMDPs) as a formalism for representing a large class of infinite-state MDPs as a (finite) set of component MDPs that may recursively call each other. The authors first present an example RMDP (a probabilistic program with holes, representing action choice points), then formally define RMDPs and their translation to infinite-state MDPs, such that each state in an RMDP corresponds to a node in the currently active component MDP, paired with the stack of component MDPs called so far.

The authors next derive a number of theoretical results about RMDPs under variants of the restriction that all policies terminate in a finite number of steps in expectation (achieved by adding discounting to an RMDP). In particular, they show that $\epsilon$-proper RMDPs are PAC-learnable, derive a generalized Bellman equation for proper RMDPs, and further derive a sound and effective continuous-space abstraction of RMDPs that preserves the optimal value function and optimal policy. In this abstraction, an RMDP state is abstracted to a tuple of a continuous vector and the node of the currently active component, enabling the application of deep reinforcement learning approaches that require finite-dimensional state encodings.

Using the above results, the authors derive a recursive Q-learning algorithm that they prove to converge in the special cases of 1-exit RMDPs and deterministic multi-exit MDPs. They also develop a deep Q-learning variant, presented in the Appendix. They then evaluate these algorithms on a number of example RMDPs across different application contexts, showing that they achieve convergence and higher total reward, unlike non-recursive (deep) Q-learning approaches.

**Questions:**

1. The following statement on Lines 54-55 initially confused me: "The 1-exit RMDPs are strictly less expressive than general RMDPs as they are equivalent to functions without any return value." It took me a while to realize that having 1-exit means you can only return a constant value, which is hence equivalent to not having a return value at all. Perhaps this can be clarified?

2. Upon seeing Figure 2, it seemed to me that RMDPs are very similar to policy program sketches for programmable reinforcement learning agents. What is the relation between these two formalisms?

3. The explanation of context-free reward machines, especially the battery example, was confusing to me (I still don't understand it). Relatedly, it was not obvious to me how the palindrome gridworld can be represented as an RMDP. I think it would be good to show the formal translation of MDPs with context-free reward machines to RMDPs somewhere in the Appendix,

4. Until I read Lines 150 to 164, I was confused by what a full state RMDP actually is. I think it'd be good to mention somewhere earlier, e.g., during the informal introduction, that the full state of an RMDP just corresponds to the current state of the active component MDP, conjoined with the stack of all component MDPs called so far. This also makes it easier to see how RMDPs have an infinite state space.

5. In lines 165 to 170, the language of "strategies" (and also "runs", later on) was unfamiliar to me -- it seems like this terminology is more common in some subfields, whereas "policy" is more standard in others. I think it would be good to briefly explain that strategies are just policies that may be history-dependent.

6. When defining the optimality equations, the terms $Ex$ and $En$ are not defined anywhere earlier in the paper. I assumed that $Ex$ is just the union of all exit nodes across component MDPs, but it's not defined on Line 137.

7. The notation for the equations defining $\text{OPT}_\texttt{cont}(M)$ was confusing. First, I was confused by the fact that $v$ seems to have varying dimension depending on what component you're in, because on the second branch, it says that $q \in Ex$ rather than $q \in Ex_i$. The notation used for the first branch is also hard to understand -- I think more explanation below in English would be helpful here.

8. Given that a *continuous* space abstraction is used for (tabular) recursive Q-learning, isn't it actually impossible to implement using a Q-table? I understand that Algorithm 2 for single-exit RMDPs avoids this, because you don't actually need to store the exit values anywhere, but for the general case, discretization is required. What impact does this have on convergence?

9. Relatedly, the Cloud Computing example is a stochastic multi-exit RMDP not covered by any of the theoretical results. Yet Algorithm 1 seems to attain convergence to the optimum, despite the lack of theoretical guarantee, and despite the use of discretization. How was this possible, and how was the true optimal strategy found and justified?

Minor comments:
- Line 66: "taks" should be task.

**Limitations:**

The authors have discussed limitations of their approach in the assumptions they make for their theoretical results. That said, there are a few questions re discretization for tabular learning that I think should also be answered or at least mentioned, as noted above.

**Strengths And Weaknesses:**

This was a really interesting (albeit theoretically dense) paper that introduces a set of representational and algorithmic ideas that are (to my knowledge) original, are likely to be of interest and significance to a number of communities who might wish to apply reinforcement learning to environments that exhibit recursive dynamics (probabilistic programs, environments with non-Markovian task rewards, etc.) By taking the general principle of the "infinite use of finite means" and applying it to MDPs, RMDPs extend the space of sequential decision problems that can be succinctly modeled.

Beyond that representational insight, the authors derive a number of useful theoretical results that can be applied to a large class of RMDPs (most of the restrictions are satisfied by the common modeling assumption of a discount factor), and further show that RMDPs admit a finite-dimensional continuous-space abstraction, thereby deriving the nice result that many RMDPs can be optimally solved by finite policies that do not depend on the state of the call-stack or state history, while also enabling the use of deep RL approaches. While the derived Q-learning algorithms unfortunately are not proven to converge in a large number of other cases (multi-exit RMDPs), they empirically appear to outperform non-recursive algorithms even in those settings. Even if the authors do not prove everything one might like, all in all this seems to me like a good start for a paper that also introduces the concept of RMDPs.

My main piece of feedback is that the exposition and presentation was unclear or confusing to me at points. The example and applications at the start helped, but at least a number of statements later on that seemed obvious to the authors were not at all clear to me. I'll highlight these in my questions and suggestions below. (Due to time constraints, I'll also note that I only briefly skimmed the proofs in the Appendix.)

Separately, in discussing related work, I was surprised by the lack of discussion of programmable reinforcement learning agents [1], which provides a formalism for potentially recursive policies, specified as subroutines with choice points that may call each other. While recursive policies are not the same thing as recursive MDPs, it stuck me that they may be similar or equivalent in some contexts, especially in the Cloud Computing / probabilistic program synthesis example, where the structure of a policy program sketch corresponds exactly to the structure of the RMDP itself. I think it would be good to cite work in this field, and discuss at least some of these connections.

Apart from that, I enjoyed reading this paper, and I think a number of research communities will benefit from its publication.

---

> ### Author Response · Authors · 2022-08-02
> **Reply to Reviewer dojD**
>
> Thank you for your careful review and helpful comments. We will implement your feedback in the final version of this paper.
>
> Answers to questions:
> 1. Your understanding is correct. We will clarify this in the final version.
>
> 2. Programmable Reinforcement Learning Agents [1] introduce domain knowledge by constraining the policy space to be a specified partial program. This view considers bounded recursion in policy space rather than our perspective of unbounded recursion in environment space. The orthogonality of these ideas means they are complementary — one can consider applying the ideas of [1] to find a policy in an RMDP. To resolve the choice points, the authors of [1] compose the partial program with the finite MDP to be solved to produce a semi-Markov decision process (SMDP). For this resulting SMDP to be finite, the partial program is assumed to not contain unbounded recursion (from [1] “Formally, the program can be viewed as a finite state machine with state space $\Theta$ (consisting of the stack, heap, and program pointer).”) This is an important distinction from our work, as we consider unbounded recursion.
>
> 3. We agree with this point and will add further explanation in the final version. The extension here is to allow a reward machine (a finite state machine with reward) to use an unbounded stack. This allows properties specified as pushdown automata to be composed with an MDP to produce an RMDP.
>
> 4. We will add this to the final version.
>
> 5. Our intended meaning of “policies” and “strategies” are identical. They are synonyms for the way in which the agent decides to select actions, and may depend arbitrarily on the history. We then quantify the restrictions to policy classes later, e.g. “positional” “stackless”. We will modify our language in the final version to use “policy” exclusively.
>
> 6. This is correct. We will add this description to line 137.
>
> 7. We will improve the presentation of this section in the final version.
>
> 8. Yes, this understanding is correct. In the general case there are two approaches for dealing with the continuous space abstraction, discretization (applied to the “Cloud” example) and deep neural network approximation (applied to the “Palindrome” example). Due to the undecidability of finding optimal policies for general finite RMDPs, these methods may not converge. However, we found that these techniques were effective empirically, and they were developed from sound principles. The “Spelunking” example is a 1-exit RMDP for which Algorithm 2 was used. We will expand this discussion in the final version.
>
> 9. The optimal policy and value for this example was determined by hand analysis. The empirical convergence to the globally optimal policy on this example is likely due to its simplicity.
>
> [1] State Abstraction for Programmable Reinforcement Learning Agents. David Andre and Stuart J. Russel. AAAI. 2002.

---

> > ### Comment · Reviewer_dojD · 2022-08-08
> > **Response to Authors**
> >
> > Thank you to the authors for their detailed replies. I appreciate the explanation of the connection between RMDPs and programmable RL agents, and I agree with your summary and the stated differences - I hadn't previously realized that they only considered policies with bounded recursion. The answers to Questions 8 and 9 have also addressed my confusions.

---

### Meta-Review · Area_Chair_5Tc9 · 2022-08-24

**Recommendation:** Accept
**Confidence:** Certain

**Metareview:**

The paper has generated a lot of discussion, and on balance the reviewers appreciate its technical contributions but find that the paper would benefit from a more in-depth discussion of its relationship to hierarchical RL.

**Award:**

No

---

### Decision · Program_Chairs · 2022-09-14

Accept